# The Role of Endoscopic Ultrasonography (EUS) in Metastatic Tumors in the Pancreas: 10 Years of Experience from a Single High-Volume Center

**DOI:** 10.3390/diagnostics14121250

**Published:** 2024-06-13

**Authors:** Alessandro Aversano, Laura Lissandrini, Daniele Macor, Martina Carbone, Sara Cassarano, Marco Marino, Mauro Giuffrè, Alessandro De Pellegrin, Giovanni Terrosu, Debora Berretti

**Affiliations:** 1Gastroenterology Unit, University Hospital of Udine, 33100 Udine, Italy; 2Department of Internal Medicine (Digestive Diseases), Yale School of Medicine, New Haven, CT 06511, USA; 3Pathological Anatomy, University Hospital of Udine, 33100 Udine, Italy; 4General Surgery Clinic and Liver Transplant Center, University Hospital of Udine, 33100 Udine, Italy

**Keywords:** EUS, pancreas, cancer, metastases, oncology

## Abstract

Background: Metastatic pancreatic lesions (MPLs) are relatively uncommon, constituting 2 to 5% of all pancreatic tumors. They often manifest as solitary lesions without distinct clinical symptoms, usually identified incidentally during radiologic imaging for the surveillance of prior malignancies. Differentiating these lesions from primary pancreatic tumors presents a significant challenge due to their nonspecific presentation. Methods: We aimed to prospectively assess the effectiveness of endoscopic ultrasound (EUS) and EUS-guided fine needle aspiration/biopsy (EUS-FNA/B) in diagnosing MPLs in a carefully selected cohort of patients presenting with pancreatic masses. Additionally, we sought to examine the relevance of specific EUS findings in supporting the initial diagnosis of MPLs and their agreement with the definitive cytological diagnosis. This study retrospectively analyzed data from 41 patients diagnosed with MPLs between 2013 and 2023, focusing on their clinical and pathological characteristics, the echogenic features of the pancreatic lesions, and the techniques used for tissue acquisition. Results: The incidence of MPLs in our cohort was 3.53%, with the most frequent primary tumors originating in the kidney (43.90%), colorectum (9.76%), lung (9.76%), lymphoma (9.76%), and breast (4.88%). MPLs typically presented as hypoechoic, oval-shaped lesions with well-defined borders and were predominantly hypervascular. Interestingly, 68.29% of the cases were discovered incidentally during follow-up of the primary tumors, while the involvement of the common bile duct was uncommon (19.51%). Conclusions: EUS and EUS-FNA/B have been validated as valuable diagnostic tools for identifying MPLs. While our findings are promising, further multicenter studies are necessary to corroborate these results and elucidate the predictive value of specific EUS characteristics in determining the metastatic origin of pancreatic lesions.

## 1. Introduction

Metastatic pancreatic lesions (MPLs) are relatively infrequent, accounting for 2% of pancreatic cancer cases. However, autoptic studies have demonstrated higher MPL prevalence, ranging from 3 to 12% of all pancreatic malignancies, with only 2% of MPLs being limited to the pancreas [1,2,3,4]. In addition, MPLs can rarely present as isolated metastatic tumors [5,6,7].

The pancreas can be a metastatic site for almost all types of tumors [1,3,6,8,9,10,11,12,13,14]. In fact, for individuals with a current or past medical history of cancer involving the kidney, skin, lung, colon, or breast, the potential for MPLs should not be overlooked [15], with the prognosis being influenced by the primary malignancy and the available treatment options [5]. 

MPLs are often asymptomatic, discovered accidentally during follow-up exams, presenting as synchronous (<3 months after first cancer diagnosis) or metachronous (>3–12 months after first cancer diagnosis). Occasionally, MPLs can be detected before the primary malignancy site is identified [1,6,16,17].

Regardless of the presentation and primary malignancy, a solid mass involving the pancreas can be characterized with endoscopic ultrasound-guided fine needle aspiration/biopsy (EUS-FNA/B), followed by cytological and immunohistochemical analysis. These are crucial to identifying the primary tumor site [6,13,18], given that endosonographic findings typically do not allow differentiation between a primary or a metastatic tumor [6]. In particular, performing FNA/B is the recommended diagnostic strategy [3,19,20] and drastically increases the diagnostic sensitivity in up to 85% of cases [18,21,22].

The primary aim of this study is to evaluate the effectiveness of EUS-FNA/B in diagnosing MPLs within a high-volume center’s select patient cohort. Secondly, we aim to characterize the clinical, pathological, and echo-endoscopic features of MPLs. These objectives are critical, given the diagnostic challenges and prognostic implications associated with MPLs, aiming to improve the EUS characterization of MPLs. 

## 2. Materials and Methods

In this retrospective cohort study, we enrolled consecutive patients referred to the Gastroenterology Service (Udine University Hospital) to undergo diagnostic evaluation with endoscopic ultrasound with fine needle aspiration/biopsy (EUSFNA/FNB) and with suspected MPL over a 10-year period (from 1 January 2013 to 31 December 2023). The study was conducted in accordance with the ethical principles for medical research involving human subjects as indicated by the Declaration of Helsinki.

### 2.1. Inclusion and Exclusion Criteria

Patients were identified using the local Anatomical Pathology System (APSyS SIO TRK ver. AXFR9.07.00.00—Insiel S.p.a., Udine, Italy), a management query software used by the Pathology Department. In our query, we included patients referred by and to the Gastroenterology Service, in possession of a cytology sample and involving the keyword “metastatic tumor to the pancreas”.

The inclusion criteria were designed to obtain a group of patients with a confirmed diagnosis of secondary tumor of the pancreas and included in adult patients (≥18 years of age) cytological and histological findings consistent with the diagnosis of metastasis to the pancreas and the pancreatic lesion, verified both radiologically and through endoscopic ultrasound study. 

Patients were categorically excluded if they exhibited a lower likelihood of having a metastasis to the pancreas: located outside the pancreas (stomach, duodenum, jejunum, lymph nodes) or other cytological/histological diagnosis (pancreatic neuroendocrine tumor). 

### 2.2. Patients and EUS Data

For all the patients, we collected the following parameters.

Personal data: sex (male or female), date of birth, date of diagnosis, age at secondary tumor diagnosis, age at primary tumor diagnosis, year of diagnosis, death (yes or no), date of death, age at death, follow-up period (expressed in months).Pathological anatomy data: adequacy of sampling, immunohistochemistry, FNA diagnosis.EUS data: pancreatic localization (head, uncinate process, neck, body, tail), size (expressed in mm), number of lesions (when multiple pancreatic lesions were found, the outcome of EUSFNA/FNB was related to the largest one), size, shape, margins, echogenicity, vascularization, elastography (expressed in KPa).Surgery data: surgery (yes or no), date of surgery, histological/immunohistochemical agreement between the surgical specimen and needle aspiration.Onset symptoms.

The EUS procedures were performed by experienced endoscopists under conscious or deep sedation using a conventional linear scope (Pentax EG3870UTK or Pentax EG38-J10UT; Pentax Precision Instruments, Orangeberg, NY, USA). Moreover, 22- or 25-gauge needles (Slimline Expect™, Boston Scientific Corp., Marlborough, MA, USA or SharkCore; Covidien, Dublin, Leinster, Ireland) were used to provide the cytological/histological samples.

### 2.3. Pathology Evaluation

The cytological samples obtained by EUS-FNA/FNB were examined on conventional PAP smears (including pathology on-site examination, when available) and on hematoxylin and Eosin slides prepared by formalin-fixed and paraffin-embedded cellblocks. In all the samples, additional immunohistochemistry tests were performed, using organ-specific markers as diagnostic tool, as detailed below:CD10, RCC and PAX8 for kidney carcinoma;CK20 and CDX2 for colorectum carcinoma;TTF1, Napsin-A and P40 for non-small-cell lung cancer and TTF1, Synaptophysin and Chromogranin-A for small-cell lung cancer;S0X10 and MART1 for cutaneous melanoma;CD10, CD20, CD23, BCL2, BCL6, PAX5 for lymphoma;OCH1E5, Glypican-3 and Arginase-1 for hepatocellular carcinoma.

In the case of metastatic chondrosarcoma, poorly differentiated medium-size cell neoplasia, showing only patchy and mild positivity for CD56, negativity for all the epithelial markers, together with the cytomorphology and oncological history, was used to establish a diagnosis.

After proper formalin fixation, the neoplastic cellularity in each sample demonstrated adequate antigenic preservation, ensuring an optimal immunohistochemical slide quality and, consequently, a certain diagnosis.

Moreover, when available for on-site or remote (by OCUS^®^40: a microscope scanner, Grundium OCUS40 Single Slide Scanner with X line Objectives (MGU-00004)—EVIDENT Europe GmbH, Hamburg, Germany) cytology interpretation, an attending pathologist made an extemporaneous assessment of the adequacy of the cytological material obtained by EUS-FNA/FNB.

### 2.4. Statistical Analysis

In this study, continuous variables are presented as the median and interquartile range (IQR), while categorical variables are expressed as the frequency and percentage. To assess the differences between deceased and alive patients, the Mann–Whitney U test was utilized for the continuous variables, and the Chi-square test was employed for the categorical variables. We arbitrarily set a survival cut-off of 1 year and performed a univariate analysis of all the variables using logistic regression. The results are reported as the odds ratios (ORs) with 95% confidence intervals (CIs) and *p*-values. A two-tailed *p*-value of less than 0.05 was considered statistically significant in all the analyses.

The statistical analysis was performed using the statistical software package Excel (version 16.43), IBM SPSS (version 29.0) and R (version 4.2.2).

## 3. Results

### 3.1. Population Baseline Characteristics (see Table 1)

In the period between 2013 and 2023, 1161 consecutive pancreatic EUS-FNA/B procedures were performed. As regards metastases to the pancreas, 41 patients were identified, of whom 25 were males (60.98%) and 16 females (39.02%). Their median age at diagnosis was 71.53 years (IQR 11.26), range 30–85 years. The median follow-up time was 24.07 months (IQR 42.64) and 25 patients (60.98%) died during this time. Furthermore, nine patients (21.95%) underwent pancreatic surgery. The surgical procedure varied based on the location of the lesions: four patients underwent a distal pancreatectomy, two patients underwent a pancreaticoduodenectomy, and in the other three patients, a total splenopancreasectomy was performed. All of these patients previously underwent nephrectomy.

Three patients underwent stereotactic body radiation therapy (SBRT) for the treatment of the metastases: one patient with metastases from chondrosarcoma, one with metastases from gastric adenocarcinoma, and the other one from renal cell carcinoma. 

In our study, we recorded an incidence of 3.53% in patients with pancreatic mass who underwent EUS-FNA/B.

The most common primary tumor was renal cell carcinoma (RCC), particularly the clear cell histological subtype, which was identified in 18 patients (43.90%). The other primary tumors were colorectal cancer (n = 4), lung cancer (both NSCLC and SCLC, n = 4), non-Hodgkin lymphoma (n = 4), breast cancer (n = 2), melanoma (n = 2), hepatocellular carcinoma (n = 2), neuroendocrine tumors (n = 2), gastric diffuse-type adenocarcinoma (n = 1), chondrosarcoma (n = 1) and uterine leiomyosarcoma (n = 1).

Carcinoma was the main histologic type of cancer metastatic to the pancreas (70.73%), followed by non-Hodgkin lymphoma (9.75%) and neuroendocrine tumor (7.31%), with others being under-represented.

Through the FNA procedure, we were able to identify 13/18 patients with RCC, 4/4 with colorectal cancer, 4/4 with lung cancer, 4/4 with non-Hodgkin lymphoma, 2/2 with breast cancer, 1/2 with melanoma, 2/2 with hepatocellular carcinoma, 1/1 with gastric adenocarcinoma, 1/1 with chondrosarcoma, and 1/1 with uterine leiomyosarcoma. Two lesions were correctly identified as small bowel NETs (neuroendocrine tumors), while another patient was diagnosed with an NET through FNA; however, on the surgical specimen, he was subsequently diagnosed with RCC. Three samples were suspicious for malignancy, while two samples were inadequate for diagnosis. Among the four remaining patients with RCC, two were diagnosed based on an operative specimen, one based on the clinical history and EUS appearance, and the last one was diagnosed after excluding a pancreatic NET through immunohistochemical and morpho-functional (PET), always taking into account the patient’s clinical history. Similarly, the other patient with metastases from melanoma, who had a specimen suggestive of neoplasm but non-diagnostic, was diagnosed based on the clinical history.

Most patients (73.17%) also had other metastatic locations. The most frequent sites were the liver (n = 14), lung (n = 14) and lymph nodes (10). The other metastatic sites were the adrenal gland (n = 8), bone (n = 7), skin (n = 6), peritoneum (n = 5), brain (n = 4), mediastinum (n = 2), kidney (n = 2), ear (n = 2), breast (n = 1), bowel (n = 1) and spleen (n = 1).

The median time between diagnosis and the discovery of metastases was 34 months (IQR 141); in 14 patients, the lesions were identified at the same time as the primary tumor, while in 26 patients, they were metachronous. Some of them were identified many years after the diagnosis of the primary tumor; in particular, one patient with RCC had a recurrence after 24 years. 

In most cases (68.29%), the patients were asymptomatic and diagnosed incidentally during the follow-up or staging of the primary tumor. Eight patients showed jaundice, two had imaging evidence of thrombosis, two patients had melaena, and one patient had pancreatic-like pain.

None of the patients had complications after EUS.

Additional gastrointestinal procedures were performed in 15 patients (35.59%): 5 patients repeated the FNA due to inadequate material for diagnosis (riFNA, 12.20%), 4 patients underwent upper endoscopy for melaena (9.76%); in order to treat jaundice, 2 endoscopic retrograde cholangiopancreatography (ERCP, 2.44%), 2 endoscopic ultrasound-guided biliary drainage (EUS-BD, 4.88%) and 1 percutaneous transhepatic biliary drainage (PTBD, 2.44%) were performed. Lastly, we performed one EUS-guided fiducial marker placement in preparation for subsequent radiotherapy.

### 3.2. Lesion Baseline Characteristics (see Table 2)

Overall, we found 75 pancreatic lesions: 26 patients had a single lesion, while the other 15 had multiple locations. The median size of the lesions was 30 mm (IQR 27), with the largest being 120 mm.

Most patients had several pancreatic locations simultaneously (12, 29.27%), where the preferred site was the head of the pancreas (n = 10, 24.39%), followed by the body (n = 7, 17.07%), tail (n = 5, 12.20%), neck (n = 5, 12.20%) and uncinated process (n = 2, 4.88%).

Immunohistochemistry was performed to allow for a better evaluation of 36 samples in order to find the origin of the metastases.

As regards the EUS features, 22 lesions were oval (54.66%) and 19 were roundish. Almost all the lesions were hypoechogenic (97.56%) and only one was hyperechogenic. The echogenicity pattern was homogeneous in 24 cases and heterogeneous in 17 cases. Most lesions had well-defined borders (60.98%), while 16 lesions had irregular borders. The vascular evaluation showed 17 hypervascular lesions, 12 lesions were moderately vascularized, 10 were avascular, and in 2 cases, the vascular pattern was not reported. 

In 32 cases, elastography was performed, in which most lesions were hard (n = 24, 58.54%), 5 were heterogeneous (12.19%) and 3 were soft (7.32%).

### 3.3. Sampling Technical Features (see Table 3)

EUS-guided fine needle sampling was successfully performed in 40 patients. In one patient, due to the infiltration of the duodenal wall, a biopsy forceps was used. Among the 40 patients, a 25-gauge needle was used in 33 cases and a 22-gauge needle was used in the remaining 7. In most cases, three (41.46%) or two (31.71%) passes were performed; in seven cases, four passes; in two cases, five passes; and in one case, six passes were performed. The preferred technique was the slow pull (78.05%), while in seven cases, suction was performed, and in one case, forceps biopsy was required. 

In 24 cases, rapid on-site evaluation (ROSE) was available, and in one case, macroscopic on-site evaluation (MOSE) was reported.

Afterwards, we considered the main continuous and categorical variables as categorized by major and minor survival at 1 year (see Table 4).

We determined through the Mann–Whitney U test that there was a statistically significant difference (*p*-value 0.026481) between the time to diagnosis of the primary tumor and the onset of metastasis compared with major and minor survival at 1 year. Hence, patients with a longer latency between diagnosis of the primary and the onset of metastasis had a longer survival. Finally, among the categorical variables, we found no statistically significant differences.

## 4. Discussion

Metastatic involvement of the pancreas may be included in the differential diagnosis of other malignant tumors to characterize suspected pancreatic focal masses. Although rare, this diagnosis should always be excluded, especially in the context of a previous history of malignancies. Recently, EUS, successfully implemented with FNA, has acquired an increasingly recognized role in the diagnostic work-up of these lesions [21,23,24]. 

Nonetheless, only a few studies, mainly case reports and case series, have attempted to systematically evaluate the significance of EUS-FNA as a diagnostic tool and prognostic predictor in this setting (see Table 5).

For the first time, Palazzo et al. [11] compared the endosonographic features of MPLs to primary pancreatic carcinoma. They reviewed the results of 7000 biliopancreatic ultrasound endoscopies performed at their tertiary center between January 1989 and July 1993; only seven were related to lesions proven histologically to be metastases to the pancreas. The authors concluded that the EUS findings of a rounded, well-delineated mass with a homogeneous isoechoic or slightly hypoechoic structure, although not specific, should strongly suggest the diagnosis of MPLs. However, these results were based on the retrospective review of EUS examinations, the study cohort was small, and the role of FNA in obtaining the histological confirmation of MPLs was not investigated as the diagnosis of metastases to the pancreas was proven histologically based on surgical specimens or CT-guided biopsies.

More recently, Fritscher-Ravens et al. [18], by retrospectively comparing the EUS findings of 114 patients with final diagnoses of primary malignancy, MPLs or focal pancreatitis, identified some EUS morphologic features concerning the texture, cystic components or vessels that might be used to differentiate between pancreatic lesions. However, they concluded that no statistically significant differences existed, making EUS appearance alone not diagnostic for pancreatic metastases. On the contrary, the authors suggested the potential advantage of cytological analysis by FNA to obtain a tissue confirmation of the metastatic origin, reaching values for the sensitivity, specificity, and accuracy of 88%, 100%, and 92%, respectively. However, the sample size was small (only 12 pancreatic lesions were found to be metastases), limiting the relevance of this study.

In two relevant retrospective cases of pancreatic metastases (28 and 24, respectively) undergoing EUS-FNA, some morphologic EUS features statistically correlated with the diagnoses of MP. In particular, De Witt et al. [23] proved that MPLs were more likely to have regular margins compared with primary cancer, while no statistically significant differences concerning the tumor size, echogenicity, consistency, or location emerged between the primary and metastatic lesions. According to Hijioka et al., the presence of well-defined borders, the absence of retention cysts, and the absence of main pancreatic duct dilation were independently predictive of MPLs. These EUS findings should increase the suspicion of MPLs, and FNA, providing samples for cytopathologic analysis, as proven to be a valuable tool for confirming the diagnoses [25].

A recent study published by Spadaccini et al. in 2023 [26] described 205 lesions with a confirmed diagnosis of MP. The EUS features of these lesions were predominantly solitary, hypoechoic, and hypervascular, with a heterogeneous pattern and well-defined borders. However, only metastases from kidney carcinoma had a stable EUS pattern (hypoechoic, hypervascular lesions with well-defined borders). The EUS-FNA/B had an overall accuracy of 97.7%, so this study proved that EUS with tissue acquisition can provide a differential diagnosis in pancreatic masses. Moreover, Spadaccini et al. did not find a specific, prevalent distribution of metastases to the pancreas to distinguish it from primary pancreatic cancer. However, in other studies, the pancreatic head was the preferred site for MPLs [11]. This is another reason why using EUS-FNA/B is essential to reach a definitive diagnosis and decide collegially on the best possible treatment for these patients.

The aforementioned studies have several limitations, given their retrospective nature or relatively small sample size. To the best of our knowledge, our study is one of the few that systematically investigate the type and frequencies of pancreatic metastases in a large cohort of consecutive patients referred with instrumental evidence of suspicious pancreatic lesions. We showed that the overall incidence of MPLs was 3.53%, approximately comparable to the literature [4,27].

Carcinoma was the prevalent histotype (70.73%), followed by non-Hodgkin lymphoma (9.75%), melanoma (4.88%), and neuroendocrine tumor (4.88%), with others under-represented. The most common primary tumor sites were the kidney and the colorectum, followed by the lung. These phenotypes resemble data from previous case reports and case series, where the kidney and carcinoma represent the primary source and the most common histological type of MP, respectively [8,15,28,29].

Finally, we found that some EUS findings, concerning the echogenicity pattern, shape, borders, vascular pattern, and others, tend to be similar in metastatic lesions. These findings make their differential diagnosis difficult.

In particular, MPLs most often showed hypoechoic echogenicity (97.56%) and appeared hypervascular (41.46%) or moderate vascular (29.27%). The morphology was mainly oval (53.66%) or roundish (46.34%), with well-defined margins, and they appeared hard (58.54%) or heterogeneous (12.19%) on elastography.

As found in others studies, we also detected some differences in the EUS findings between pancreatic adenocarcinoma (PA) and MPLs. Compared to metastatic lesions, the EUS of a PA typically reveals a solid mass, distinctly hypoechoic relative to the adjacent pancreatic tissue. The lesion often shows that the borders with the surrounding parenchyma are not that neat, manifesting the early tendency of the PA to invade nearby structures. In a Doppler study, the lesion generally appears non-vascularized [30]. Elastosonographic evaluation usually shows a harder lesion, while the rest of the pancreatic parenchyma appears normal [31]. In contrast, the MPL appearance on EUS is closer to neuroendocrine tumors (pNETs), which appear as solid hypoechoic oval lesions, with well-defined borders and a hypervascular pattern (see Figure 1) [32]. 

In conclusion, EUS-FNA has been confirmed as a useful diagnostic tool in the setting of MP, with 86% diagnostic accuracy. Only seven cases were without a certainty diagnosis. Specifically, in 2 cases out of 41, the cytology was inconclusive, in 3 cases out of 41, the cytology was only suggestive of malignancy, and in only 1 case, the diagnosis was wrong (pNET vs. kidney cancer). 

Another crucial point of our study was that most patients (73.17%) also had other metastatic locations, mainly involving the liver, lungs, and lymph nodes. This restricted the possibility of a surgical approach, for which a survival benefit has been demonstrated in patients with isolated RCC metastases to the pancreas [33]. In fact, only nine patients diagnosed with metastases from RCC underwent pancreatectomy. Of those nine patients, eight were alive at the time chosen as the end of follow-up. Nevertheless, the patient who died before the end of this time had a survival of 92 months. 

On the other hand, three more patients underwent SBRT for the treatment of the metastases. The patient with chondrosarcoma was still alive at the time chosen as the end of follow-up, while the one with gastric adenocarcinoma survived 4 months after SBRT. The last one, who suffered RCC, underwent radiation therapy for palliation because he developed abdominal pain, melaena and anemia. Although these results do not seem encouraging, our sample is very limited and further studies are needed to understand the role of SBRT. 

Our study had several limitations. First, it was a retrospective study conducted at a single facility without a comparison group. In addition, the study only included patients with a positive diagnosis of MPLs and did not include a control group. Comparing the demographic and clinical features as well as the EUS findings in the control group could have contributed more significance to our results. Second, only a few patients underwent surgery. Thus, the diagnosis of MPLs was based on EUS-FNA cytology alone in almost all cases.

## 5. Conclusions

Metastases to the pancreas are a rare event in the natural history of several cancers; they can occur even long after the primary tumor diagnosis. Hence, their accurate detection can change the treatment approach, and EUS-guided tissue acquisition with FNA/B plays a key role in the diagnostic work-up.

Moreover, the EUS features suggest that some of them may be highly predictive of secondary tumors of the pancreas.

Finally, although from the partial data presented in this paper the SBRT does not seem to be very beneficial, we point out that such a conclusion would be difficult to draw from the small sample we considered in our work, as the attention paid to SBRT was incidental with respect to the main focus of this work. We conclude that further studies should be conducted in order to obtain a deeper understanding of the actual benefits of SBRT. 

From this perspective, these results might encourage further and more extensive multicenter studies to better clarify these aspects.

## Figures and Tables

**Figure 1 diagnostics-14-01250-f001:**
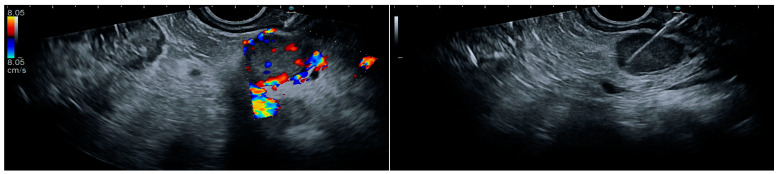
Typical EUS features of MP from kidney carcinoma: hypoechoic, hypervascular lesions, with well-defined borders.

**Table 1 diagnostics-14-01250-t001:** Population baseline characteristics.

**Patients, n**		**41**	
Male, n (%)		25 (60.98%)	
Median age at diagnosis in years, (IQR)		71.53 (11.26)	
Deaths, n (%)		25 (60.98%)	
Median follow-up in months, (IQR)		24.07 (42.64)	
Surgery, n (%)		9 (21.95%)	
Stereotactic body radiation therapy (SBRT), n (%)		3 (7.31%)	
**Primary tumor**	**n (%)**	**Diagnosis FNA/B**	**n (%)**
Kidney	18 (43.90%)	Kidney	13 (31.71%)
Colorectal	4 (9.76%)	Colorectal	4 (9.76%)
Lung	4 (9.76%)	Lung	4 (9.76%)
Lymphoma	4 (9.76%)	Lymphoma	4 (9.76%)
Breast	2 (4.88%)	Breast	2 (4.88%)
Melanoma	2 (4.88%)	Melanoma	1 (2.44%)
Liver	2 (4.88%)	Liver	2 (4.88%)
Neuroendocrine tumor	2 (4.88%)	Neuroendocrine tumor	3 (7.32%)
Stomach	1 (2.44%)	Stomach	1 (2.44%)
Chondrosarcoma	1 (2.44%)	Chondrosarcoma	1 (2.44%)
Uterine leiomyosarcoma	1 (2.44%)	Uterine leiomyosarcoma	1 (2.44%)
Unknown	0	Suspect for neoplasia	3 (7.32%)
		Inadequate sample	2 (4.88%)
**Other metastatic location, n (%)**		**30 (73.17%)**	
Liver		14 (34.15%)	
Lung		14 (34.15%)	
Lymph node		10 (24.39%)	
Adrenal gland		8 (19.51%)	
Bone		7 (17.07%)	
Skin		6 (14.63%)	
Peritoneum		5 (12.20%)	
Brain		4 (9.76%)	
Mediastinum		2 (4.88%)	
Kidney		2 (4.88%)	
Ear		2 (4.88%)	
Breast		1 (2.44%)	
Bowel		1 (2.44%)	
Spleen		1 (2.44%)	
**Median time between diagnosis and metastasis in months, (IQR)**		34 (141)	
**Diagnostic timing, n (%)**			
Synchronous		14 (34.15%)	
Metachronous		27 (65.85%)	
**Signs and symptoms, n (%)**			
Asymptomatic		28 (68.29%)	
Jaundice		8 (19.51%)	
Imaging thrombosis		2 (4.88%)	
Melaena		2 (4.88%)	
Pancreatic-like pain		1 (2.44%)	
**Post-EUS complications, n**		0	
**Additional gastrointestinal procedures, n (%)**		15 (35.59%)	
None, n (%)		26 (63.41%)	
RiFNA, n (%)		5 (12.20%)	
Upper endoscopy, n (%)		4 (9.76%)	
EUS-BD, n (%)		2 (4.88%)	
ERCP, n (%)		2 (2.44%)	
EUS-guided fiducial marker placement, n (%)		1 (2.44%)	
PTBD, n (%)		1 (2.44%)	

EUS: Endoscopic ultra-sonography; EUS-BD: EUS-biliary drainage; ERCP: endoscopic retrograde cholangiopancreatography; PTBD: percutaneous biliary drainage.

**Table 2 diagnostics-14-01250-t002:** Lesion baseline characteristics.

**Lesions, n**	**75**
Single, n (%)	26 (63.41%)
Multiple, n (%)	15 (36.59%)
Median size in mm (IQR)	30 (27)
Immunohistochemistry, n (%)	36 (87.80%)
**Anatomical location, n (%)**	
Uncinated process	2 (4.88%)
Head	10 (24.39%)
Neck	5 (12.20%)
Body	7 (17.07%)
Tail	5 (12.20%)
Several locations simultaneously	12 (29.27%)
**EUS features**	
*Shape*, n (%)	
Roundish	19 (46.34%)
Oval	22 (53.66%)
*Echogenicity*, n (%)	
Hypoechogenic	40 (97.56%)
Hyperechogenic	1 (2.44%)
*Echogenicity pattern*, n (%)	
Homogeneous	24 (58.54%)
Heterogeneous	17 (41.46%)
*Borders*, n (%)	
Regular	25 (60.98%)
Irregular	16 (39.02%)
*Vascularization*, n (%)	
Avascular	10 (24.39%)
Moderate	12 (29.27%)
Hypervascular	17 (41.46%)
Unreported	2 (4.88%)
**Elastography, n (%)**	32 (78.05%)
Hard	24 (58.54%)
Heterogeneous	5 (12.19%)
Soft	3 (7.32%)

**Table 3 diagnostics-14-01250-t003:** Sampling technical features.

**FNA/B needle, n (%)**	
25 G	33 (80.49%)
22 G	7 (17.07%)
**Number of passes, n (%)**	
Unreported	1 (2.44%)
2	13 (31.71%)
3	17 (41.46%)
4	7 (17.07%)
5	2 (4.88%)
6	1 (2.44%)
**Aspiration technique, n (%)**	
Suction	7 (17.07%)
Slow pull	32 (78.05%)
Forceps biopsy	2 (4.88%)
**ROSE (rapid on-site evaluation), n (%)**	24 (58.54%)
**MOSE (macroscopic on-site evaluation), n (%)**	1 (2.44%)

**Table 4 diagnostics-14-01250-t004:** Main variables categorized by major and minor survival at 1 year.

	Survival < 1 Year	Survival > 1 Year
**Main continuous variables**	Median (IQR)	Median (IQR)
Age at diagnosis in years, (IQR)	74.35 (5.44)	70.89 (10.85)
Time between diagnosis and metastasis in months, (IQR)	2.00 (62.50)	62.00 (118.00)
Size in mm, (IQR)	30.00 (21.75)	30 (30.00)
**Main categorical variables**	Count (Percentage)	Count (Percentage)
Male	8 (57.14%)	17 (62.96%)
Female	6 (42.86%)	10 (37.94%)
Surgery	0	9 (33.34%)
Non-surgery	14 (100%)	18 (66.67%)
Other metastatic location	11 (78.57%)	19 (70.37%)
Non-other metastatic location	3 (21.43%)	8 (29.63%)
Single lesion	10 (71.43%)	16 (59.26%)
Multiple lesions	4 (28.57%)	11 (40.74%)
Synchronous	9 (64.29%)	5 (18.52%)
Metachronous	5 (35.71%)	22 (81.48%)

**Table 5 diagnostics-14-01250-t005:** Comparison of EUS characteristics in patients with MP. Abbreviations: NA = Not available; ML = Main lesion (i.e., data only available for the main lesion).

	Palazzo et al., 1996 [11]	Fritscher-Ravens et al., 2001 [18]	DeWitt et al., 2005 [23]	Hijioka et al., 2011 [25]	Spadaccini et al., 2023 [26]	Current Study
**N. Patients**	7/7000	12/114	24/37	28	101	41/1161
** Tumor location **						
**Head**	5	10	15	9	57	10
**Uncinated process**	NA	NA	NA	NA	43	2
**Neck**	0	0	0	0	37	5
**Body**	0	2	5	13	50	7
**Tail**	1	0	3	3	49	5
**Several locations simultaneously**	1	0	1	3	NA	12
**Tumor size (mm; mean ± range)**	40 (15–60)	28 (18–40)	36 (16–70)	34.4 (13–90)	25.4 (±15.2)	30 ** (27) **
** N. of lesions **	16	12	29		205	75
**Single**	6	12	22	23	59	26
**Multiple**	1 (10)	0	2	5	42	15
** Echogenicity **				NA	ML	
**Isoechoic**	15	0	0		2	0
**Hypoechoic**	1	12	20		85	40
**Hyperechoic**	0	0	1		3	1
**Anechoic**	0	0	1		0	0
**Mixed**	0	0	2		0	0
** Echo pattern **			NA		ML	
**Homogeneous**	15	2		14	40	24
**Heterogeneous**	1	10		14	50	17
** Borders **		NA				
**Regular**	15		13	18	64	25
**Irregular**	1		11	10	26	16

Data not available; ** Median (IQR); Data only available for main lesions.

## Data Availability

The data from all the subjects involved in the study are available upon request.

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
