# Peer review of "The Role of Endoscopic Ultrasonography (EUS) in Metastatic Tumors in the Pancreas: 10 Years of Experience from a Single High-Volume Center"

_diagnostics, 2024, doi:10.3390/diagnostics14121250_

Round 1

Reviewer 1 Report

Comments and Suggestions for Authors

In the current manuscript, the authors have retrospectively analyzed data from 41 patients with MPLs, focusing on their clinical and pathological characteristics, the echogenic features of the pancreatic lesions, and the 27 techniques used for tissue acquisition. The authors demonstrated not only the clinicopathological features but also the important role of EUS-guided tissue acquisition in diagnosing MPLs.

However, the novelty of this study is low in my opinion. Authors may enhance the quality of their study by presenting data that can inform the development of treatment strategies for MPLs. For example, data on the treatment of MPLs following diagnosis.

Author Response

Thank you very much for taking time to review this manuscript. 

I agree with you. Initially I decided to exclude these data because I thought they were not enough. I will try to add them to my manuscript.

Reviewer 2 Report

Comments and Suggestions for Authors

The manuscript is regarding the characteristics of tumors metastatic to the pancreas diagnosed via EUS. The study aim may not be novel, but it remains valuable. However, several points need attention:

1.      suggest the authors to provide the institution review board certification.

2.      The tables need to be revised. The authors may revised according to the table forms in the journal.

3.      The authors may revised the text of manuscript thoroughly, such as the word ”cytology” in page 3 line 113 is not adequate.

4.      A paragraph about the comparison of primary pancreas cancer and metastatic to pancreas cancer may be added. The authors may consider comparison these two type of lesions in their series, but this is not mandatory.

Comments on the Quality of English Language

minor

Author Response

Thank you very much for taking time to review this manuscript. Please find the detailed responses below and the corresponding revisions/corrections highlighted.

  1. I do not understand where I have to fill "the institution review board certification" in. I have already filled it in at the end of the manuscript, before the reference (section: Institutional Review Board Statement).
  2. I think I have correctly revised all the tables with the journal format.
  3. I do not understand the reason why the words: "cytological/histological samples" are wrong.
  4. Thank you again for the advice, I have just added a paragraph about the comparison between primary pancreas cancer and metastatic tumors to the pancreas.

Round 2

Reviewer 1 Report

Comments and Suggestions for Authors

I appreciate your appropriate corrections to my review comments. I believe the addition of information on post-diagnosis treatment has enhanced the paper's appeal and utility in real-world clinical practice for readers.